# To Learn or Not to Learn, That is the Question — A Feature-Task Dual Learning Model of Perceptual Learning

Xiao Liu[1,2], Muyang Lyu[1,2], Cong Yu[2*], and Si Wu[1,2*]

[1]Peking-Tsinghua Center for Life Sciences, Academy for Advanced Interdisciplinary Studies,
Beijing Key Laboratory of Behavior and Mental Health,
IDG/McGovern Institute for Brain Research,
Center of Quantitative Biology, Peking University
[2]School of Psychological and Cognitive Sciences, Peking University
{xiaoliu23, yucong, siwu}@pku.edu.cn
lyumuyang@stu.pku.edu.cn

## Abstract

Perceptual learning refers to the practices through which participants learn to improve their performance in perceiving sensory stimuli. Two seemingly conflicting phenomena of specificity and transfer have been widely observed in perceptual learning. Here, we propose a dual-learning model to reconcile these two phenomena. The model consists of two learning processes. One is task-based learning, which is fast and enables the brain to adapt to a task rapidly by using existing feature representations. The other is feature-based learning, which is slow and enables the brain to improve feature representations to match the statistical change of the environment. Associated with different training paradigms, the interactions between these two learning processes induce the rich phenomena of perceptual learning. Specifically, in the training paradigm where the same stimulus condition is presented excessively, feature-based learning is triggered, which incurs specificity, while in the paradigm where the stimulus condition varies during the training, task-based learning dominates to induce the transfer effect. As the number of training sessions under the same stimulus condition increases, a transition from transfer to specificity occurs. We demonstrate that the dual-learning model can account for both the specificity and transfer phenomena observed in classical psychophysical experiments. We hope that this study gives us insight into understanding how the brain balances the accomplishment of a new task and the consumption of learning effort.

## 1 Introduction

Perceptual learning refers to the practices through which human participants learn to improve their performances of perceiving sensory stimuli [1]. Perceptual learning has been widely used as a research paradigm to ascertain the learning strategies in the brain. Over the years, two seemingly contradictory phenomena have been widely found in perceptual learning. One is specificity, referring to that after learning, the improved performance of participants is restricted to the trained location or the trained feature of the stimulus [2, 3]. The other is transfer, referring to that after learning, the improved performance is transferable to untrained locations or untrained features of the stimulus [4,

---

*These authors contributed equally to this work and should be considered co-corresponding authors.

38th Conference on Neural Information Processing Systems (NeurIPS 2024).

5, 6]. In early experiments in which participants were typically trained excessively under the same stimulus condition (e.g., either at the same retinal location or presenting the same stimulus feature), specificity was predominantly found, and stimulus features used in experiments include contrast [7, 8], orientation [9, 10, 11, 12], spatial frequency [13, 11, 12, 14], motion direction [15, 16, 17], and retinal location [18, 19, 20, 14]. Thus, once in a while, specificity was regarded as a hallmark distinguishing perceptual learning from other learning types [3]. Later on, enriched training paradigms were used, and it was found that perceptual learning can also exhibit the dominating transfer effect. For instances, by manipulating the task difficulty [21, 22, 23, 24], the training intensity [25, 26, 27], or the training protocol [28, 29, 30, 31, 32, 33, 34, 35, 36], the originally observed specificity-dominating effect become transfer-dominating.

**Related works.** A large volume of computational models has been proposed in the literature to unveil the mechanism of perceptual learning. Since specificity was predominantly reported in early experiments, early computational models mainly focused on exploring the neural mechanism underlying specificity. For instance, Poggio et al. proposed the hyper basis function model (HyperBF) [37], which considers that the enhanced perception performance comes from the changes of basis functions or connection weights of neurons; Teich and Qian proposed a model which considers that the enhanced performance comes from the changes of tuning curves of low-level neurons [38]; Petrov et al. proposed the reweighting model, which considers that the enhanced performance comes from the read-out weights from feature representations to decision neurons [39, 40]. All these models *implicitly* assume that perceptual learning only incurs local changes in the network, and hence, the learning effect is localized and not transferable to other parts of the neural system needed for processing untrained conditions. Later on, motivated by the observation of transfer, new computational models focusing on the transfer mechanism were proposed. For examples, Dosher et al. refined their reweighting model by including an extra layer with network-wide connections to implement translation-invariant feature representation, which achieves a certain level of transfer effect [41]; Solgi et al. introduced an extra off-line learning procedure which generalizes the local learning result to untrained locations/features, making the learning performance transferable [42]; Li et al. employed a convolutional neural network (CNN) to simulate perceptual learning [43], utilizing that the CNN implements translation-invariant feature extraction, and hence their model can explain some transfer phenomena. While these models may solve the transfer problem, they fail to account for specificity, which was also vividly observed in experiments. Overall, perceptual learning still lacks a unified computational model accounting for both specificity and transfer that have been observed in different experiments.

In this study, we propose a novel computational model to reconcile the seemingly conflicting phenomena of specificity and transfer in perceptual learning. Our model is based on the fact that in the experiments observing specificity, participants were often excessively trained under the same stimulus condition (either at the same retinal location or presenting the same stimulus feature); while in the experiments observing transfer, the stimulus condition was varied during the training (either the location or the stimulus feature). This indicates that excessive training of the same stimulus condition is critical to induce specificity. To account for both specificity and transfer arising from different training paradigms, we propose a dual-learning framework for perceptual learning in the brain. The framework consists of two intertwined learning processes: one is task-based, and the other is feature-based. Specifically, task-based learning aims to quickly adapt to a new task by utilizing existing neural representations of the external world. It enables the brain to master new skills quickly and is transferable to untrained conditions. On the other hand, feature-based learning aims to refine neuronal representations to reflect the statistical changes of the external world. It enables the brain to represent the external world faithfully and it only takes effect when the external environment is substantially changed. We argue that in the experiments exhibiting transfer, task-based learning dominates; while in the experiments exhibiting specificity, feature-based learning dominates, and the latter is triggered by the neural system being excessively exposed to the same stimulus condition, triggering feature-based learning to catch up the statistical change of the external environment. Since featured-based learning leads to localized changes in feature representations adhered to the trained condition, its effect is not transferable to untrained conditions.

To substantialize the dual-learning framework, we built a hierarchical neural network model, which consists of three sequential stages of information processing responsible for, respectively, feature extraction, feature-based learning, and task-based learning. Specifically, we use a classical basis function network [37] to implement feature extraction representing preliminary image processing, a

feedforward network with unsupervised Hebbian learning to implement feature-based learning, and a CNN with global max pooling to represent task-based learning. We set the rate of feature-based learning to be much smaller than that of task-based learning, such that task-based learning will dominate initially in training, and feature-based learning will take effect after the same stimulus condition is presented excessively. We demonstrate that the dual-learning model can well explain the specificity and transfer phenomena observed in classical psychophysical experiments.

## 2 Specificity vs. transfer in perceptual learning

Various tasks, ranging from the discrimination of basic visual features such as Vernier or orientation to the recognition of visual images such as motion or face, have been applied to study perceptual learning. These experiments unveiled three generic characteristics of perceptual learning, including specificity, transfer, and a transition from transfer to specificity when the number of training sessions under the same stimulus condition increases. They are reviewed below.

We use a Vernier discrimination task to introduce the specificity effect [28]. In this task, participants learned to discern which of two vertical orientations was more leftward (or equivalently rightward) (Fig. 1A(i)). The stimuli were always presented at the same retinal location. Through intensive training, participants progressively improved their sensitivity to the offset between two orientations until reaching a threshold (the blue dashed line in Fig. 1A(ii)). After training, it was found that participants could not generalize their enhanced perceptual performances to untrained retinal locations, displaying the effect of specificity (red squares in Fig. 1A(ii)).

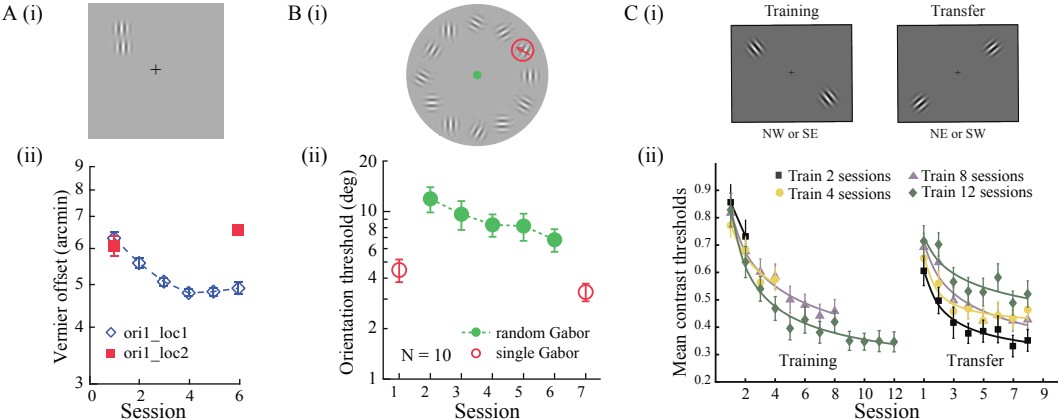

Figure 1: Specificity vs. transfer in perceptual learning. A. An example of specificity, adapted from [28]. (i) A Vernier discrimination task: discerning the offset between two vertical orientations. (ii) The threshold of detectable offset at the training retinal location decreased with training sessions (blue diamonds), in comparison to pre-/post-training thresholds at an untrained retinal location (red squares). B. An example of transfer, adapted from [34]. (i) An orientation discrimination task: Gabor stimuli varied across 47 location/orientation conditions were used for training. The red circle and arrow highlight the location/orientation not included in training but tested pre-/post-training. (ii) The orientation discrimination threshold with random Gabors over 47 stimulus conditions decreased with training sessions (green circles) in comparison to pre-/post-training thresholds at an untrained condition (red circles). C. Transition from transfer to specificity, adapted from [26]. (i) An orientation discrimination task: retinal locations and orientations of visual stimuli used in training and transfer assessment. (ii) Left panel: the contrast sensitivity threshold decreased with training sessions (black, yellow, purple, and green lines denoting 2, 4, 8, and 12 numbers of training sessions, respectively). Right panel: the transfer effect decreased with the number of training sessions.

We use an orientation discrimination task to introduce the transfer effect [34]. In this task, participants learned to distinguish which of two successively presented orientations was more clockwise (or equivalently more anticlockwise) (Fig. 1B(i)). By training with varied combinations of retinal location and orientation, participants progressively improved their sensitivity to the offset between two orientations until reaching a threshold (the green dashed line in Fig. 1B(ii)). After training, it

was found that participants could generalize their enhanced perceptual performances to untrained conditions, displaying the effect of transfer (red circles in Fig. 1B(ii)).

The above two experiments highlight the importance of the number of training sessions under the same stimulus condition in inducing the specificity or the transfer effect. It is expected that with the increased number of training sessions under the same condition, the learning effect should go from transfer to specificity gradually. Indeed, this was confirmed experimentally in an orientation discrimination task [26]. In this task, participants underwent different numbers of training sessions under the same condition, followed by a switch to untrained conditions to measure the extent of transfer (Fig. 1C(i)). It was found that with the increased number of training sessions under the same condition, the perceptual performance of participants was improved (the left panel of Fig. 1C(ii)), whereas the transfer effect was decreased (the right panel of Fig. 1C(ii))

## 3  The dual-learning model

To reconcile the phenomena of specificity and transfer in perceptual learning, we propose a dual-learning model. As depicted in Fig. 2, the model consists of three sequential information processing stages, which are feature extraction, feature-based learning, and task-based learning. We use the Vernier discrimination task as an example to introduce the model.

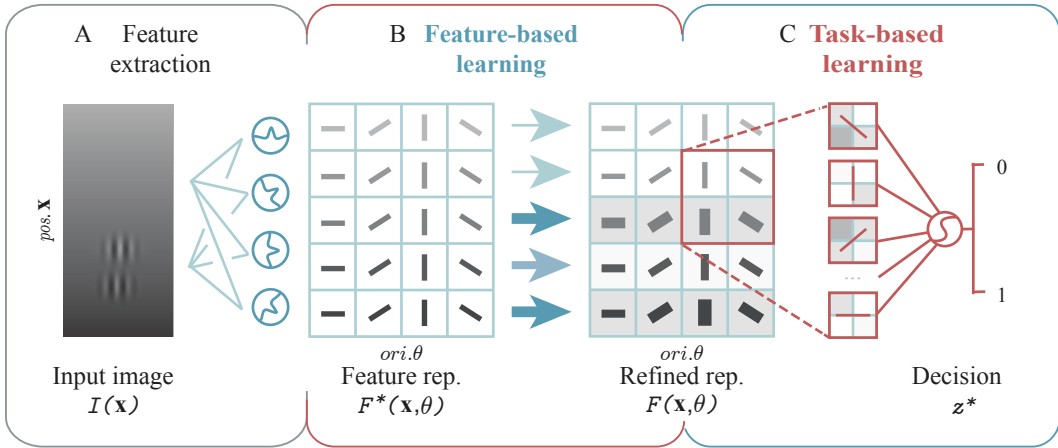

Figure 2: Overview of the dual-learning model. The Vernier discrimination task is used as an example. A. Feature extraction. It involves using basis functions to transform an image $I(\mathbf{x})$ into feature representations $F_t^*(\mathbf{x}, \theta)$, where $\mathbf{x}$ and $\theta$ denote the position and orientation features. B. Feature-based learning. It refines feature representations to $F_t(\mathbf{x}, \theta)$ to reflect the statistical changes of external inputs. The feedforward connections are updated following the Hebbian learning rule, and they are strengthened at locations where stimuli are presented excessively, inducing location-specific changes in feature representations. C. Task-based learning. Using convolutional layers and global max pooling, it integrates the task-relevant information from feature representations $F_t(\mathbf{x}, \theta)$ to make the decision $z^*$.

### 3.1  The model structure

**Feature Extraction.**   This models the functions of the retina, the Lateral Geniculate Nucleus (LGN), and the input layer of V1 in the visual pathway, extracting preliminary features from input images to form retinotopic feature representations (see Fig. 2A). There is no plasticity at this stage, and we employ the HyperBF network [37] to model feature extraction. Denote $I_t(\mathbf{x})$ the input image presented at trial $t$ and $F_t^*(\mathbf{x}, \theta)$ the extracted feature representation, which is expressed as:

$$F_t^*(\mathbf{x}, \theta) = \text{norm}\left[G(\mathbf{x} - \mathbf{x}', \theta) * I_t(\mathbf{x}')\right], \tag{1}$$

where $\mathbf{x}$ and $\theta$ denote the position and orientation feature in the image, respectively. The variable $\mathbf{x}'$ represents the position of neighboring pixels, used in the convolution operation. The Gabor function $G(\mathbf{x} - \mathbf{x}', \theta) = 1/2\exp\left[-(\mathbf{x} - \mathbf{x}')^2/(2\sigma_\mathbf{g}^2)\right]\cos\left[2\pi\mathbf{x}'/\lambda\cos(\theta) + \psi\right]$, with $\sigma_g$ the standard

deviation of the Gaussian envelope, $\lambda$ the spatial frequency, and $\psi$ the phase offset (for details, see Sec. A.1 in the Appendix). The symbol $*$ represents the convolution operation and the symbol $\mathrm{norm}(\cdot)$ denotes a normalization operation, i.e., $\mathrm{norm}(F^*) = (F^* - F^*_{\mathrm{mean}})/F^*_{\mathrm{std}}$, with $F^*_{\mathrm{mean}}$ and $F^*_{\mathrm{std}}$ the mean and standard deviation of the representation variable $F^*$.

**Feature-Based Learning.** This models the plasticity in the early visual cortex, which refines feature representations to capture the statistical change of the external environment (see Fig. 2B). For simplicity, we employ a feedforward network with unsupervised Hebbian learning to implement this process. Denote $F_t(\mathbf{x}, \theta)$ the refined feature representations at trial $t$, which is given by

$$F_t(\mathbf{x}, \theta) = \mathrm{norm}\left[\sum_{\mathbf{x}'} W_t(\mathbf{x}, \mathbf{x}') * F_t^*(\mathbf{x}', \theta)\right], \tag{2}$$

where $W_t(\mathbf{x}, \mathbf{x}')$ represents the feedforward connection of the network (Fig. 2B). $W_t(\mathbf{x}, \mathbf{x}')$ is updated following the competitive Hebbian learning rule, which is expressed as

$$W_{t+1}(\mathbf{x}, \mathbf{x}') = W_t(\mathbf{x}, \mathbf{x}') + \eta \Delta W_t(\mathbf{x}, \mathbf{x}'), \tag{3}$$

with $\eta$ as the learning rate. The detail of $\Delta W_t(\mathbf{x}, \mathbf{x}')$ is presented in Sec. A.2 in the Appendix. This learning rule enhances neuronal representations at locations where the stimulus is frequently presented, thereby inducing location-specific changes in feature representations.

**Task-Based Learning.** This models the information read-out process in the higher visual cortex (see Fig. 2C). We use a small convolutional network with three convolutional layers, followed by global max pooling, to implement this process. Denote $z_t^*$ the decision at trial $t$, which takes a value of $(0, 1)$ representing the upper orientation is more leftward or more rightward, respectively. During decision-making, if the value is greater than $0.5$, it is classified as $1$; if it is less than $0.5$, it is classified as $0$. Its value is calculated by,

$$z_t^* = \mathrm{sigmoid}\{\max\{\Phi[F_t(\mathbf{x}, \theta)]\}\}, \tag{4}$$

where $\Phi(\cdot)$ represents the multi-layer convolution operation that integrates task-related information from the feature representation $F_t(\mathbf{x}, \theta)$. The $\max(\cdot)$ function indicates the max pooling operation, which realizes translation-invariant computation and supports transferable learning. The sigmoid function, expressed as $\mathrm{sigmoid}(x) = 1/\left[1 + \exp(x)\right]$ converts the output of the max pooling into a probability, representing the final decision. The true decision at each trial is denoted as $z_t$, taking only binary values $0$ or $1$. Task-based learning updates the parameters in $\Phi(\cdot)$ by minimizing the cross entropy loss, $L_t = -\left[z_t \ln(z_t^*) + (1 - z_t)\ln(1 - z_t^*)\right]$, and backpropagation is used (for details, see Sec. A.3 in the Appendix).

### 3.2 The interplay between feature- and task-based learning

We used a Vernier discrimination task to analyze the properties of the dual-learning model. The model outputs $0$ if the upper orientation is more leftward compared to the lower one and outputs $1$ otherwise. We trained the model at a fixed location and then tested its performances at three untrained locations having increasing distances to the trained one (Fig. 3A). For the details, see Sec. B in the Appendix. We conducted the below ablation studies.

First, we froze feature-based learning and only enabled task-based learning. As shown in Fig. 3B, the discrimination accuracies at both trained and untrained locations increase gradually with train epochs, indicating that task-based learning facilitates transfer over locations. This property comes from the fact that task-based learning employs a maxing pooling operation, which is translation-invariant.

Second, we froze task-based learning and only enabled feature-based learning. The weights of the readout layer were initialized after completing task-based learning, as described above. As shown in Fig. 3C, We see that the discrimination accuracy at the trained location keeps a relatively high value, whereas the accuracies at untrained locations drop quickly to a nearly chance level (50%). This demonstrates that feature-based learning induces specificity, reinforcing the model's performance at the trained location while diminishing its transfer to untrained locations. To display the location-specific change in feature representation, we calculated neural representation changes at different locations during the training. As shown in Fig. 3D, we see that as the training goes on, the neural representation at the trained location remains stable (high similarity), while the neural

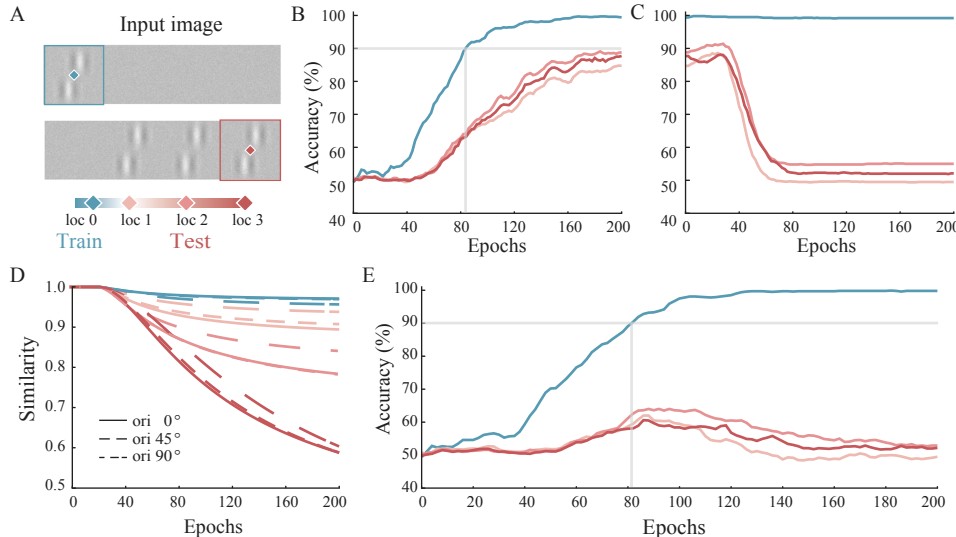

Figure 3: Properties of the dual-learning model. A. A Vernier discrimination task. Upper: the training stimulus. Lower: testing stimuli at three untrained locations. B. Discrimination accuracy vs. training epochs with only task-based learning on. It displays transfer effects to all untrained locations. C. Discrimination accuracy vs. training epochs with only feature-based learning on, following the completion of training in panel B. The model performances at untrained locations drop dramatically, display the effect of specificity. D. Similarity of feature representations before and after feature-based learning. Solid lines represent vertically oriented ($0°$) stimuli used in training, dashed lines represent diagonally oriented ($45°$) stimuli, and dotted lines represent horizontally oriented ($90°$) stimuli. E. Discrimination accuracy vs. training epochs with both feature-based and task-based learning on. In panels B and E, the gray lines indicate the number of epochs required to reach $90\%$ accuracy. The results presented are averaged over $50$ repetitions. For more details, see Sec. B in the Appendix.

representations at untrained locations change dramatically. This indicates that feature-based learning induces location-specific changes in feature representations.

Finally, we analyzed the model performance with both feature-based and task-based learning enabled. In particular, we set the rate of feature-based learning to be relatively much smaller than that of task-based learning. Although a direct numerical comparison between the two learning rates is not meaningful due to differences in their roles and magnitudes, the chosen configuration allows us to observe a clear distinction in the dynamics of the two learning processes. As shown in Fig. 3E, we observe that: 1). The combination of feature-based and task-based learning accelerates the training process at the train location, requiring fewer epochs to reach $90\%$ accuracy compared to using task-based learning alone (gray line in Fig. 3, mean epochs: task-based learning $= 84.7$, combined feature-and task-based learning $= 81.3$). This difference is statistically significant ($t = 3.90$, $p < 0.001$), based on 50 repetitions for each condition. 2). The training promotes transfer to untrained locations initially, but as time goes on, the transfer effect decreases, while the specificity effect increases. This shift reflects that due to different learning rates, task-based learning dominates initially, which induces transfer; while feature-based learning gradually takes effect, which induces specificity.

## 4 Reproducing classical findings in perceptual learning

In this section, we used the dual-learning model to reproduce the classical findings in perceptual learning. We employed a Vernier discrimination task as an example, as this paradigm has been widely used in psychophysical experiments. In these experiments, the discrimination threshold, defined as the intensity of stimuli that participants can accurately discriminate with about an $80\%$ success rate, was used to measure the learning effect. In our simulations, we adjusted the offset between two vertical orientations in the task to create a series of difficulty levels and chose the offset value corresponding to $80\%$ model correctness as the threshold (for details, see Sec. C in the Appendix).

In the first experiment, we adopted the training paradigm similar to that in [28]. Specifically, we trained the model with fixed orientation and location (ori1_loc1) and evaluated model performances across various combinations of orientation and location before and after the training (Fig. 4A). The results are presented in Fig. 4B, which show that the model exhibits a significantly improved performance in the trained condition (ori1_loc1, blue diamonds), whereas this improvement is not transferable to untrained locations (ori1_loc2, light red squares), untrained orientations (ori2_loc1, medium red triangles), or combinations of untrained location and orientation (ori2_loc2, dark red circles). Fig. 4C further summarizes the learning improvements under different conditions, showing good agreement with the experimental data [28]. The results demonstrate that our model successfully replicates the classical specificity phenomenon observed in perceptual learning.

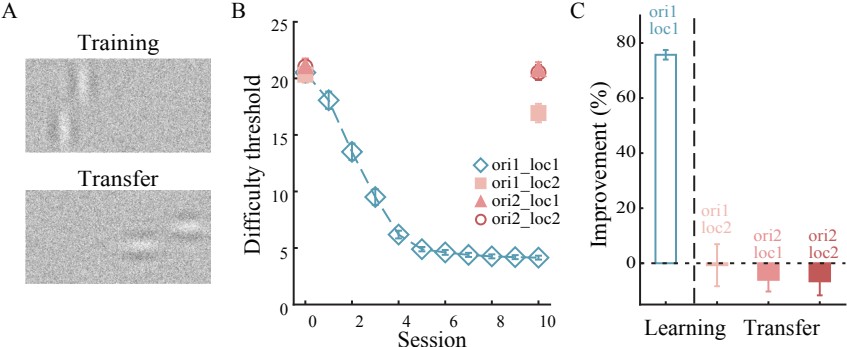

Figure 4: Specificity in perceptual learning via condition-specific training. A. A Vernier discrimination task similar to that in [28]. The visual stimulus used for training (top, loc1_ori1) and an example of stimuli with the different location and orientation for transfer evaluation (bottom, loc2_ori2). B. Learning curves and thresholds for pre- and post-testing in different conditions. The threshold at the trained condition (ori1_loc1, blue diamonds) decreases significantly after training, while thresholds at untrained conditions (ori1_loc2, light red squares; ori2_loc1, medium red triangles; ori2_loc2, dark red circles) do not exhibit significant decline. C. Statistical results of training and transfer improvements. They show substantial gain at the trained condition (ori1_loc1, blue bar) and negligible or no improvement at untrained conditions (ori1_loc2, light red; ori2_loc1, medium red; ori2_loc2, dark red).

In the second experiment, we adopted the training paradigm similar to that in [34]. Specifically, the model was trained with stimuli presented at varying locations, either in a random or sequential order, and the transfer effect was evaluated at an untrained location (Fig. 5A). The results are presented in Fig. 5B-C, showing that the model's learning effect is successfully transferred to the untrained location. Fig. 5D further summarizes the learning improvements when trained across multiple locations either randomly or sequentially. These results agree well with the experimental data [34], demonstrating that our model successfully replicates the classical transfer phenomenon in perceptual learning.

In the third experiment, we adopted a training paradigm similar to that in [26]. Specifically, the number of training sessions was varied, and the transfer effect was evaluated using a new condition that combined an untrained location and orientation (Fig. 6A). The results are shown in Fig. 6B. In the trained condition (left panel), performance thresholds gradually decreased with the increasing number of training sessions. However, in the untrained condition (right panel), the trend reversed: the more sessions completed in the trained condition, the higher thresholds were observed in the untrained condition. Fig. 6C further summarizes the progression of learning improvements during the training and transfer phases, agreeing well with the experimental data [26]. Thus, our model successfully replicates the classical phenomenon of transition from transfer to specificity with the increasing number of training sessions.

In the fourth experiment, we adopted the training paradigm called double training similar to that in [28]. In this experiment, following the first step training of the classical specificity task as in Fig. 1, we introduced second step training with stimulus at a new location and orientation. After double training, we evaluated the model's transfer performance under conditions untrained in either step. The results, shown in Fig. 7B, reveal that after double training, the original specificity effect in the

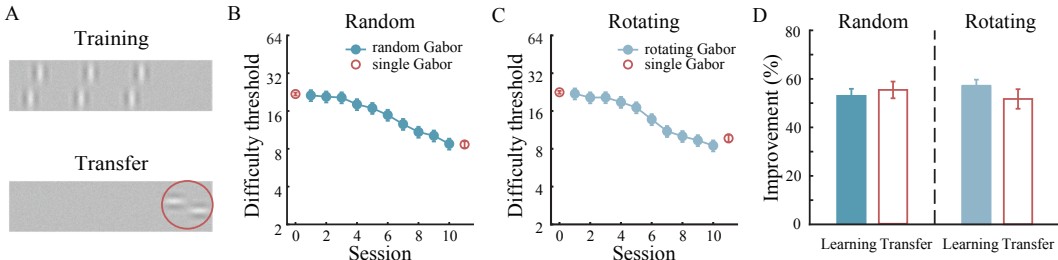

Figure 5: Transfer in perceptual learning via varied training conditions. A. A Vernier discrimination task similar to that in [34]. During training, visual stimuli were presented at three distinct locations with two orientations: horizontal and vertical (top). For transfer evaluation, a single stimulus was presented at a new, untrained location (bottom), highlighted with a red circle. B. Random training condition. The learning curve and thresholds for training across multiple conditions randomly. The threshold at the training conditions (dark blue circles) decreases significantly, and the transfer condition (red circles) shows a similar decline. C. Rotating training condition. The learning curve and thresholds for training with stimuli rotating. The threshold at the training conditions (light blue circles) decreases significantly, and the transfer condition (red circles) shows a similar decline. D. Summary of learning and transfer improvements in different conditions.

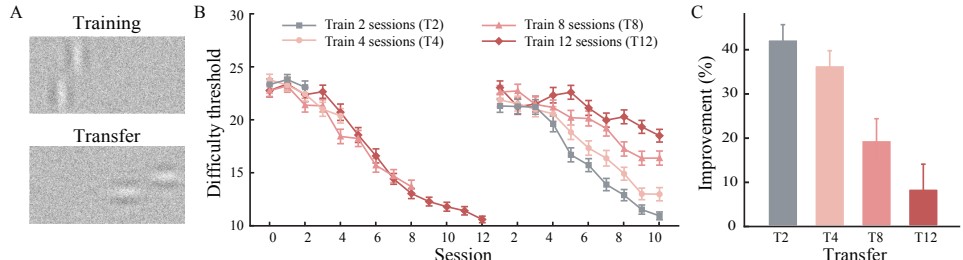

Figure 6: Transition from transfer to specificity with the increased number of training sessions. A. A Vernier discrimination task similar to that in [26]. Visual stimulus used for training (top, loc1_ori1) and stimuli for transfer evaluation (bottom, loc2_ori2). B. The learning curves under different training (left) and transfer (right) conditions. The thresholds are depicted with different colored curves representing different numbers of training sessions: gray (2 sessions), light red (4 sessions), medium red (8 sessions), and dark red (12 sessions). C. Summary of transfer improvements with varied training sessions. Each bar represents the improvement in transfer performance following 2, 4, 8, or 12 training sessions, respectively.

first step training now becomes transferable. The underlying reason is intuitively understandable. The second step training modified the feature representations adhered to the first step training, and hence reduced specificity (or equivalently increased transfer). Fig. 7C further summarizes the learning improvements across different training steps and conditions, demonstrating that our model successfully replicates the double training phenomenon in perceptual learning.

In summary, the adaptability of perceptual learning is governed by the interaction between specific feature-based learning at lower levels and transferable task-based learning at higher levels. Condition-specific training leads to the dominance of feature-based learning, resulting in significant specificity. In contrast, training with varied conditions allows task-based learning to dominate, enhancing transfer. This difference enables a transition from transfer to specificity as stimulus repetitions increase. Although not exclusively from the Vernier discrimination task, the observed changes in specificity and transfer are influenced by training paradigms rather than the tasks themselves. The simplicity of the Vernier discrimination task, requiring only a single stimulus presentation per trial, underscores its utility in illustrating these principles without the need for complex processing, making it ideal for our model demonstrations.

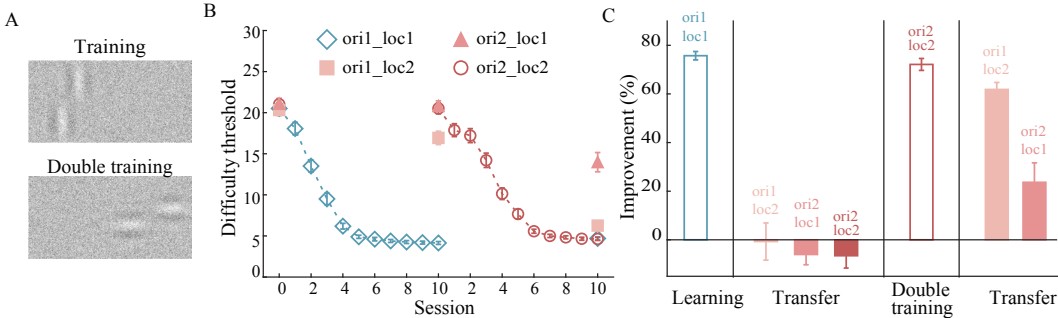

Figure 7: Transfer in perceptual learning via double training. A. A Vernier discrimination task similar to that in [28]. The visual stimulus used for the first step training (top, loc1_ori1) and the visual stimulus used for the second step training (bottom, loc2_ori2). B. Learning curves and thresholds for pre- and post-testing in different conditions. The left panel corresponds to the results of the first step training as in Fig. 4B. The right panel displays the results of double training. Notably, the threshold at the second trained condition (ori2_loc2, deep red circles) decreases significantly after double training, and thresholds at untrained locations (ori1_loc2, light red squares) or untrained orientations (ori2_loc1, medium red triangles) also exhibit notable declines. C. Statistical results of training and transfer improvements. The left panel corresponds to the results of the first step training as in Fig. 4C. The right panel displays the results of double training. A substantial gain at the second trained condition (ori2_loc2, deep red bar) and at untrained conditions (ori1_loc2, light red; ori2_loc1, medium red)

## 5 Conclusions and discussions

All learning agents, including the brain, face a fundamental challenge: balancing task performance with the cost of learning. Intuitively, if the agent does not need to account for the feature distribution in the external world, it can quickly adapt by reusing existing feature representations to complete tasks efficiently and at low cost. However, when the agent identifies meaningful statistical patterns in the environment, it can refine its feature representations to enhance precision and effectiveness. Although beneficial, this adaptation is both resource-intensive and slow, as the agent must first distinguish between genuine environmental changes and random fluctuations. In essence, to learn or not to learn, is a generic question faced by all learning agents.

In this work, starting from the goal of reconciling the conflicting phenomena of specificity and transfer in perceptual learning, we present a solution of the brain, i.e., the dual-learning framework. This framework consists of two learning processes: a task-based one and a feature-based one. Specifically, task-based learning is fast, which enables the agent to learn to accomplish a task rapidly by using existing feature representations; while feature-based learning is slow, which enables the agent to improve feature representations to reflect the statistical change of the external environment.

Our dual-learning model successfully replicates and elucidates classical experimental findings related to specificity and transfer in perceptual learning. It reveals that the interaction between the slow-changing, specific feature learning at the early visual pathway and the flexible, transferable task learning at the higher visual pathway governs the adaptability of perceptual learning. Typically, learning tends to adjust the readout of neural representations (i.e., task-based learning) rather than altering the representations themselves (i.e., feature-based learning), unless there is a significant change in the statistical properties of external information. Thus, the default state of learning favors transfer. However, in traditional experimental paradigms, the frequent repetition of stimuli leads to the dominance of feature-based learning, thereby exhibiting significant specificity; by limiting the repetition of stimuli, task-based learning can dominate, thereby enhancing transfer. As such, perceptual learning can display a transition from transfer to specificity as the number of stimulus repetitions increases.

Our dual-learning model can be regarded as a computational modeling implementation of the two-stage model theory [44]. While the two-stage model theory aims to address the contradictions between task-related and task-unrelated perceptual learning, it posits the existence of feature-based and task-based plasticity within perceptual learning. Thus, our model is a practical realization of

this theory. Furthermore, the dual-learning framework aligns with the reverse hierarchy theory [45], which suggests that learning processes invert the sequence of visual information processing from top-down. As tasks become more difficult and specific, lower levels of the neural hierarchy are increasingly engaged, aligning learning outcomes with enhanced task specificity. This alignment illustrates how the dual-learning model not only accommodates but also substantiates theoretical perspectives on perceptual learning's hierarchical nature, providing a robust framework for exploring how different learning mechanisms interact within the brain's architecture.

**Limitation and future work.** In this study, as a first step to elucidate the notion of dual-learning, we have built a very simple network model without including many biological details of the visual pathway. In future work, we will extend the current model to include more biological details and apply the model to explain more cognitive functions of the brain. In essence, the dual-learning framework reflects that the brain employs a learning strategy to balance the accomplishment of a new task and the consumption of learning effort, that is, if the statistics of the environment are unchanged, low-cost task-based learning is applied; otherwise, high-cost feature-based learning is triggered. The same balance requirement is faced by other learning agents. We, therefore, expect that the dual-learning framework has the potential to be applied in AI applications, and we will explore this issue in future work.

## Acknowledgments and Disclosure of Funding

This work was supported by the National Natural Science Foundation of China (no. T2421004, S.W.), the Science and Technology Innovation 2030-Brain Science and Brain-inspired Intelligence Project (no. 2021ZD0200204, S.W.), and the STI2030-Major Projects grant (no. 2022ZD0204600, C.Y.).

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

# Appendix

## A  Model Details

The hierarchical neural network model used in this study consists of three sequential stages:

### A.1  Feature Extraction

As defined in Eq. 1 of the main text, the feature representation $F_t^*(\mathbf{x}, \theta)$ is calculated as follows:

$$F_t^*(\mathbf{x}, \theta) \quad = \quad \mathrm{norm}\left[G(\mathbf{x} - \mathbf{x}', \theta) * I_t(\mathbf{x}')\right], \tag{A1}$$

where $G(\mathbf{x} - \mathbf{x}', \theta)$ represents the basis function, preferred for its ability to capture specific position $\mathbf{x}$ and orientation $\theta$ features from the input image. The symbol $*$ represents the convolution operation, and the $\mathrm{norm}(\cdot)$ denotes a normalization operation, i.e., $\mathrm{norm}(F^*) = (F^* - F_{\mathrm{mean}}^*)/F_{\mathrm{std}}^*$, with $F_{\mathrm{mean}}^*$ and $F_{\mathrm{std}}^*$ the mean and standard deviation of the representation variable $F^*$.

The selected basis function is a Gabor function characterized by a sinusoidal wave modulated by a Gaussian envelope. It is mathematically expressed as:

$$G(\mathbf{x} - \mathbf{x}', \theta) = G_0 \exp\left[-\frac{(\mathbf{x} - \mathbf{x}')^2}{2\sigma_g^2}\right] \cos\left[2\pi \frac{\mathbf{x}'}{\lambda} \cos(\theta) + \psi\right], \tag{A2}$$

where $\mathbf{x}$ and $\theta$ define the preferred position and orientation of the basis function.

We configured the network with $40 \times 18$ basis functions to uniformly cover all possible positions and angles. Specifically, the 40 basis functions span the positional space, evenly distributing across different position scales, while the 18 basis functions cover the angular space, with each basis function representing a 10-degree increment (i.e., $18 = 180°/10°$). It is important to note that while $\mathbf{x}$ traditionally represents a two-dimensional position within the image, we simplified the model by setting the size of the basis function to the height of the image, thus reducing the two-dimensional position variable to one dimension. This simplification does not affect the outcome of our experiments.

Table A1 below summarizes the parameters used for the Gabor function in our experiments.

Table A1: Gabor Parameters for Basis Function

| | | |
|---|---|---|
| Amplitude | $G_0$ | 0.5 |
| Spatial frequency | $\lambda$ | 0.01 cycles/pixel |
| Phase | $\psi$ | $\pi/2$ radians |
| Standard deviation | $\sigma$ | 30 pixels |

### A.2  Feature-Based Learning

As defined in Eqs. 2-3 in the main text, the refined feature representations $F_t(\mathbf{x}, \theta)$ at each trial $t$ are computed based on the feedforward connection weights $W_t(\mathbf{x}, \mathbf{x}')$. Below are the mathematical details.

#### A.2.1  Feedforward Convolution Process

The refined feature representations are updated using a convolution process that integrates feedforward connection weights with previous feature representations:

$$F_t(\mathbf{x}, \theta) \quad = \quad \mathrm{norm}\left[\sum_{\mathbf{x}'} W_t(\mathbf{x}, \mathbf{x}') * F_t^*(\mathbf{x}', \theta)\right]. \tag{A3}$$

The feedforward connection weight $W_t(\mathbf{x}, \mathbf{x}')$ from position $\mathbf{x}'$ in the previous layer to position $\mathbf{x}$ is defined as:

$$W_{t+1}(\mathbf{x}, \mathbf{x}') \quad = \quad \frac{A_{t+1}(\mathbf{x}')}{\sqrt{2\pi}a} \exp\left[-\frac{(\mathbf{x} - \mathbf{x}')^2}{2a^2}\right]. \tag{A4}$$

---

The code is available on Github at `https://github.com/XiaoLiu-git/ToLearnOrNotToLearn`.

Here, $A_{t+1}(\mathbf{x}')$ represents the magnitude of the connection, while $a$ controls the spread of the connection. For simplification, we consider $a \to 0$, the connection weight simplifies to:

$$\lim_{a \to 0} W_{t+1}(\mathbf{x}, \mathbf{x}') = \begin{cases} A_{t+1}(\mathbf{x}') & \text{if } \mathbf{x} = \mathbf{x}', \\ 0 & \text{otherwise.} \end{cases} \tag{A5}$$

This indicates that, in the limit, the connections become fully localized, creating direct, one-to-one mappings between corresponding positions.

The normalization operation $\text{norm}(\cdot)$ and the convolution operation $*$ are consistent with previous descriptions.

### A.2.2 Competitive Hebbian Learning Updating

The strength of feedforward connections, $A_t(\mathbf{x})$, is updated following competitive Hebbian learning rules [46], aimed at reinforcing location-specific changes in the neuronal representations.

First, competitive learning identifies strong responses from prior feature representations for weight updates:

$$\widetilde{F}_{t-1}(\mathbf{x}, \theta) = \text{sign}\left[F_{t-1}(\mathbf{x}, \theta) - F_0\right], \tag{A6}$$

$$\widetilde{F}_t^*(\mathbf{x}, \theta) = \text{sign}\left[F_t^*(\mathbf{x}, \theta) - F_0^*\right], \tag{A7}$$

where $F_0$ and $F_0^*$ are thresholds set to $0.6$, effectively selecting the top $20\%$ of neurons based on response strength.

Then, Hebbian learning computes and updates the connection strength $A_t(\mathbf{x})$:

$$\Delta A_t(\mathbf{x}) = \text{mean}_\theta\left[\widetilde{F}_{t-1}(\mathbf{x}, \theta)\right] \text{mean}_\theta\left[\widetilde{F}_t^*(\mathbf{x}, \theta)\right], \tag{A8}$$

$$A_{t+1}(\mathbf{x}) = A_t(\mathbf{x}) + \eta \Delta A_t(\mathbf{x}) \tag{A9}$$

where $\Delta A_t(\mathbf{x})$ represents the updated value for weights, $\eta_f$ is the learning rate, and $\text{mean}_\theta(\cdot)$ calculates the average over different orientations, considering the columnar structure of orientation preference in the primary visual cortex.

Table A2 below summarizes the parameters used for the feature-based Learning in our experiments.

Table A2: Parameters for Feature-based Learning

| Parameter | Symbol | Value |
|---|---|---|
| Threshold of $F_t(\mathbf{x}, \theta)$ | $F_0$ | 0.6 |
| Threshold of $F_t^*(\mathbf{x}, \theta)$ | $F_0^*$ | 0.6 |
| Learning rate of feature-based learning | $\eta_f$ | 0.003 |

### A.3 Task-Based Learning

As defined in Eq. 4 of the main text, the decision $z_t^*$ at each trial $t$ is computed using a convolutional network followed by global max pooling. The decision $z_t^*$ takes a value of 0 or 1, representing whether the upper orientation is more leftward (0) or more rightward (1), respectively. During decision-making, the output is classified as 1 if the value exceeds $0.5$; otherwise, it is classified as 0.

The CNN used in our model is a streamlined convolutional neural network comprising three convolutional layers.

- **Layer 1:** Inputs a single channel and uses a 3x3 convolutional kernel to output 6 channels.
- **Layer 2:** Processes the 6 channels with another 3x3 kernel, producing 10 channels.
- **Layer 3:** Compresses the 10 channels into a single output channel using a 3x3 convolution.

The network employs ReLU activation after the first two convolutional layers to introduce non-linearity, followed by flattening and a 1D max pooling on the final output to ensure a well-structured output.

The task-based learning updates the network parameters through cross-entropy loss minimization:

$$L_t = -\left[z_t \ln(z_t^*) + (1 - z_t)\ln(1 - z_t^*)\right], \tag{A10}$$

where backpropagation is used for efficient parameter adjustment with an Adam optimizer. The learning rate $\eta_t$ is set at 0.001 to ensure gradual and steady learning.

## B  Experimental Protocol for Ablation Studies

### B.1  Stimulus

The Vernier discrimination images used in this study were adapted from the Matlab code in [41]. Each image has a size of $800 \times 200$ pixels (Fig. 3A). The stimulus within each image consists of a $200 \times 200$ pixels Vernier pattern overlaid on a background of visual noise.

The Vernier stimulus consists of two Gabor patches arranged either horizontally or vertically. A single Gabor patch is defined by:

$$G_\theta = G_0 \cos[2\pi\lambda(x\cos\theta + y\sin\theta) + \psi] \exp\left(-\frac{x^2 + y^2}{2\sigma^2}\right), \tag{A11}$$

where the parameters are described in Table A3.

Table A3: Gabor Parameters for Stimulus

| Parameter | Symbol | Value |
|---|---|---|
| Amplitude | $G_0$ | 0.6 |
| Spatial frequency | $\lambda$ | 0.02 cycles/pixel |
| Orientation | $\theta$ | $0°$ or $90°$ (vertical or horizontal stimulus) |
| Phase | $\psi$ | $\pi/2$ radians |
| Standard deviation | $\sigma$ | 25 pixels |

The two Gabor patches are offset by a value $O$ that is randomly generated for each trial:

$$O = 2(2 + N_d), \tag{A12}$$
$$N_d = [n_d], \tag{A13}$$
$$n_d \sim \mathcal{N}(0, 0.5^2), \tag{A14}$$

where $[\cdot]$ denotes rounding to the nearest integer, and $n_d$ follows a normal distribution with a mean of 0 and a standard deviation of 0.5.

The background noise is generated by:

$$Noise = a_{noise} \cdot \mathcal{N}(1, 2^2), \tag{A15}$$

where $a_{noise}$ is the noise amplitude, and the normal distribution has a mean of 1 and a standard deviation of 2.

### B.2  Simulation Details

The model is trained at a fixed location and tested at three untrained locations with increasing distances from the trained location, as illustrated in Fig. 3A of the main text.

Procedure:

- Train the model for 200 epochs, using batches of 16 trials each.
- Measure the discrimination accuracy at both trained and untrained locations to assess the model's performance.
- Average all results over 50 repetitions to ensure the robust and reliable findings.

To analyze the contributions of feature-based and task-based learning, three specific ablation studies are conducted:

1. **Task-Based Learning Only:** Feature-based learning is disabled, isolating the effect of task-based learning to on model performance across different locations.

2. **Feature-Based Learning Only:** Task-based learning is disabled, and only feature-based learning operates. Initialize the weights as the weight completing task-based learning. During training, the similarity between the feature representations $F_t^*(\mathbf{x}, \theta)$ and the refined representations $F_t(\mathbf{x}, \theta)$ under different stimulus locations (1 trained location and 3 untrained locations) and orientations ($0°$, $45°$, $90°$) is measured. The similarity is calculated as:

$$Similarity = \frac{n\left(\sum F_t^* F_t\right) - \left(\sum F_t^*\right)\left(\sum F_t\right)}{\sqrt{\left[n\sum(F_t^*)^2 - \left(\sum F_t^*\right)^2\right]\left[n\sum(F_t)^2 - \left(\sum F_t\right)^2\right]}}, \quad (A16)$$

where $n$ denotes the total number of representations.

3. **Combined Feature-Based and Task-Based Learning:** Both learning modes are activated, but feature-based learning uses a lower rate(0.00001) compared to task-based learning (0.001). The model starts with weights initialized at the beginning of task-based learning, which helps to evaluate how the combined learning dynamics influence the speed and effectiveness of training across different spatial contexts.

## C   Reproduction of Experimental Phenomena

### C.1   Stimulus

In this study, Vernier discrimination stimuli were generated as described previously. The stimuli were available in two sizes: $400 \times 200$ pixels (used for the first, third, and fourth experiments; Fig. 4A, Fig. 6A and Fig. 7A) and $800 \times 200$ pixels (used for the second experiment; Fig. 5A).

Each Vernier stimulus consists of two Gabor patches with orthogonal orientations: ori1 ($\theta = 0°$; horizontal orientation) and ori2 ($\theta = 90°$; vertical orientation), as described in Eq. A11.

In the first, third, and fourth experiments, the Vernier stimulus was centered at two locations: loc1 (100 , 100) and loc2 (300, 100). In the second experiment, two additional locations were added: loc3 (500, 100) and loc4 (700, 100).

The offset $O$ between the two Gabor patches varied across 10 difficulty levels, defined as:

$$O = 2\left(2D + N_d\right), \quad (A17)$$

where $D \in [1, 10]$ is the difficulty level, $N_d = [n_d]$ is the rounded difficulty noise, and:

$$n_d \sim \mathcal{N}\left(\mu, \sigma_d^2\right), \; \sigma_d = 0.5. \quad (A18)$$

The probability density function $\psi\left(n_d\right)$ for the truncated normal distribution was as follows:

$$\psi\left(n_d\right) = \begin{cases} 0, & n_d \leq a \\ \frac{\phi\left(\mu, \sigma_d^2; n_d\right)}{\Phi\left(\mu, \sigma_d^2; b\right) - \Phi\left(\mu, \sigma_d^2; a\right)}, & a < n_d < b \\ 0, & b \leq n_d \end{cases} \quad (A19)$$

where $\phi$ and $\Phi$ were the probability density and cumulative distribution functions, respectively. The values $a = -1$ and $b = 1$ are the truncation points.

The background noise $N$ is generated as:

$$N = a_{noise} \cdot n, \; n \sim \mathcal{N}\left(\mu_n, \sigma_n^2\right) \quad (A20)$$

where $a_{noise}$ was the noise intensity, $n$ was the raw noise, and $N$ followed a normal distribution with mean $\mu_n = 1$ and variance $\sigma_n^2 = 2$. The stimulus images generated in each trial were mutually independent.

## C.2 Measuring Difficulty Threshold

All the results in this study were obtained by independently training 100 models with the same set of parameters under different experimental conditions (e.g., variations in random initialization and training stimulus noise).

To compare with human perception experiments, we measured the difficulty thresholds for each model at every test point. A threshold represents the intensity at which a difference between two stimuli is just detectable, with stimuli below it considered subliminal. Here, we used the constant stimuli method to measure the threshold at which model accuracy reached $80\%$, comparable to the $79\%$ threshold in human subject experiments[28] [26][30][34]. The constant stimuli method is a standard psychophysical approach for measuring perceptual thresholds [47], involving the presentation of stimuli at several constant levels and fitting a psychometric function to the responses.

At each test point, we evaluated model accuracy across 10 difficulty levels. Each level included 20 positive and 20 negative samples, totaling 40 trials. The model's accuracy at each level was used to fit a psychometric function based on the cumulative normal distribution:

$$A_i = f\left(D_i\right) = \left(\frac{1}{\sigma_{fit}\sqrt{2\pi}}\int_{-\infty}^{D_i}\exp\left(-\frac{(t-\mu_{fit})^2}{2\sigma_{fit}^2}\right)dt + \epsilon_i\right)(1-C) + C, \qquad \text{(A21)}$$

where $A_i$ denotes the accuracy at difficulty level $D_i$, $\mu_{fit}$ and $\sigma_{fit}$ are the mean and standard deviation of the fitted distribution, and the $\epsilon_i$ is the error between the observed and fitted accuracy. And $C$ denoted the chance level accuracy, which was $0.5$ for this binary forced-choice task.

The goal of the fitting process is to minimize the total error:

$$\min_{\mu_{fit},\sigma_{fit}} \sum_{i=1}^{n_{point}} \epsilon_i^2, \qquad \text{(A22)}$$

where $n_{point} = 10$. Once fitted, the threshold difficulty $D_{threshold}$ corresponding to an $80\%$ accuracy level calculated as:

$$D_{threshold} = f^{-1}\left(A_{threshold}\right), \; A_{threshold} = 0.8. \qquad \text{(A23)}$$

The fitting process was conducted using the `scipy.optimize.curve_fit` function from the Python `scipy` library, with upper bounds for $\mu_{fit}$ and $\sigma_{fit}$ set to 30 and 13, respectively, based on initial pre-training results. The cumulative distribution function and its inverse were computed using `scipy.stats.norm.cdf` and `scipy.stats.norm.ppf`.

In addition to directly presenting the threshold values, we also calculated the Percent Improvement (PI) for each experiment to quantify the effect of learning. The PI for model $m$, denoted as $I_m$ was defined as:

$$I_m = \frac{-(threshold_{after_m} - threshold_{before_m})}{threshold_{before_m}} \qquad \text{(A24)}$$

where $threshold_{before_m}$ and $threshold_{after_m}$ were the threshold measured before and after learning, respectively.

## C.3 Experimental Simulation

### C.3.1 First Experiment

- Training Condition: ori1_loc1
- Testing Conditions: ori1_loc2, ori2_loc1, ori2_loc2
- Stimulus Noise Intensity: $a = 0.6$
- Learning Rate: $\eta_f = 0.3$ (feature-based learning), $\eta_t = 0.001$ (task-based learning)

**Pre-training** Since the model also needed to measure the threshold before formal training (i.e., pre-testing), at this time, due to the random initialization of the model parameters, the stimulation accuracy rate of all difficulties was the chance level (i.e., $50\%$), and the psychometric function was a horizontal line, which could not measure the threshold. Therefore, the model needed to be pre-trained

before formal training. During pre-training, the model was trained with tasks set at difficulty level 10 (the easiest) of all training and transfer conditions. Each condition consisted of 16 trials, forming one training batch. Stimuli of all conditions were sequentially trained batch by batch for 20 epochs. Stimuli of the two orientations trained a shared feature-based learning model component and two distinct task-based model components.

**Formal Training**    The formal training was conducted under the training condition at difficulty level 10. Every 16 trials formed a training batch, and 20 training batches constituted one training session. During the training process, both the feature-based learning model component and the task-based model component corresponding to that orientation were trained. After completing training across 10 training sessions, the traditional training phase was completed.

**Pre- and Post-Testing**    We performed pre- and post-testing on all training and transfer conditions, before the formal training (after pre-training) and after the formal training, respectively. We measured the $80\%$ threshold of all the training and transfer conditions and calculated the improvement using the method mentioned in Sec. C.2.

### C.3.2    Second Experiment

- Training Conditions: ori1_loc2, ori1_loc3, ori1_loc4, ori2_loc1, ori2_loc2, ori2_loc3, ori2_loc4
- Testing Condition: ori1_loc1
- Stimulus Noise Intensity: $a = 0.6$
- Learning Rate: $\eta_f = 0.3$ (feature-based learning), $\eta_t = 0.001$ (task-based learning)

**Pre-training**    Same as Sec. C.3.1.

**Formal Training**    For the random training condition, the trials were shuffled, while for the rotation condition, trials were presented sequentially in dictionary order. Other procedures were the same as Sec. C.3.1.

**Pre- and Post-Testing**    Testing followed the same protocol as described in Sec. C.3.1.

### C.3.3    Third Experiment

- Training Condition: ori1_loc1
- Transfer Condition: ori2_loc2
- Stimulus Noise Intensity: $a = 1.2$
- Learning Rate: $\eta_f = 3$ (feature-based learning), $\eta_t = 0.001$ (task-based learning)

**Pre-training**    Same as Sec. C.3.1.

**Formal Training**    For the first training phase, the model was trained on the training condition for 2, 4, 8, or 12 sessions according to different conditions. For the transfer training phase, models of all the conditions were trained on the transfer condition for 10 sessions. Other details were similar to Sec. C.3.1.

**Pre- and Post-Testing**    Testing followed the same protocol as in Sec. C.3.1.

**Statistical Comparison**    In Fig. 6B, we argue that at the final transfer session, the thresholds are from low to high as the training sessions increase. The results of the t-tests conducted on the data are presented below in Table A4, which contains the p-values of T-tests for points at the final transfer session in Fig. 6B. As observed, the statistical differences between any two conditions of the experiment are significant. This finding aligns with the results depicted in Fig. 1C(ii), indicating the consistency across different parts of the study.

Table A4: Statistical Comparison of the Third Experiment

|      | T2      | T4      | T8      | T12     |
|------|---------|---------|---------|---------|
| T2   | 1       | 0.0054  | 2e-11   | 2.1e-21 |
| T4   | 0.0054  | 1       | 0.00026 | 1.4e-09 |
| T8   | 2e-11   | 0.00026 | 1       | 0.02    |
| T12  | 2.1e-21 | 1.4e-09 | 0.02    | 1       |

### C.3.4 Fourth Experiment

- Training Condition: ori1_loc1
- Double Training Condition: ori2_loc2
- Testing Conditions: ori1_loc2, ori2_loc1
- Stimulus Noise Intensity: $a = 1.2$
- Learning Rate: $\eta_f = 3$ (feature-based learning), $\eta_t = 0.001$ (task-based learning)

**Pre-training**  Same as Section C.3.1.

**Formal Training**  The model was trained for 10 sessions under both the initial and double training conditions. Other details were similar to Sec. C.3.1.

**Pre- and Post-Testing**  Pre- and post-testing followed the same protocol as in Sec. C.3.1.

**Outlier Removal**  Some results shown in Fig. 7B and Fig. 7C appear inconsistent, which may cause confusion (e.g. the threshold of ori1_loc2 at session 10 in the training condition is lower than that at session 0 in Fig. 7B, indicating an improvement in perception. However, in Fig. 7C, the improvement of ori1_loc2 after the first step training is around 0, suggesting no learning). This inconsistency arises due to that Fig. 7B presents the average of 100 simulation runs, while the improvement in Fig. 7C was calculated by averaging the rate of improvement from each simulation, where a few outliers affected the results. After excluding 4 outliers (with z-scores of improvements greater than 4), the results are consistent (see Fig. A1).

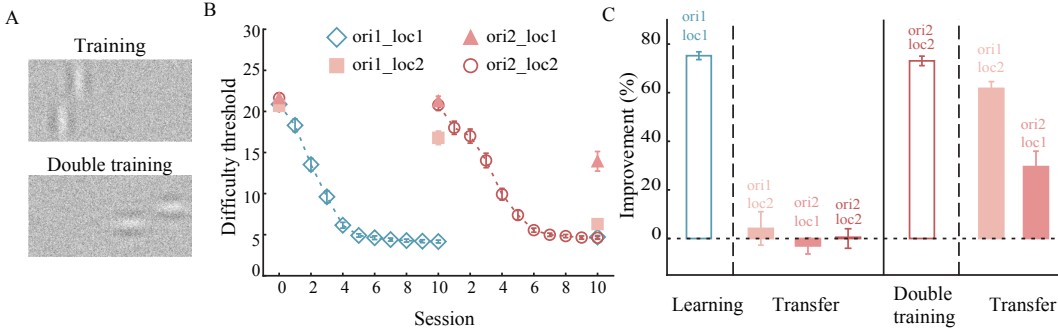

Figure A1: Transfer in perceptual learning via double training after removing four sets of outliers. The detailed figure description can be found in Fig. 7.

### C.4 Additional Ablation Study

We conducted two additional ablation studies to highlight the importance of the relative speeds of feature-based and task-based learning. The results, based on 100 simulations, confirm that both slow feature-based learning and fast task-based learning are essential for replicating perceptual learning phenomena in our model.

### C.4.1 Accelerating Feature-based Learning

As shown in Fig. A2, we increased the learning rate of feature-based learning by tenfold and replicated the four experiments from Sec. 2 in the main text. The detailed results are as follows:

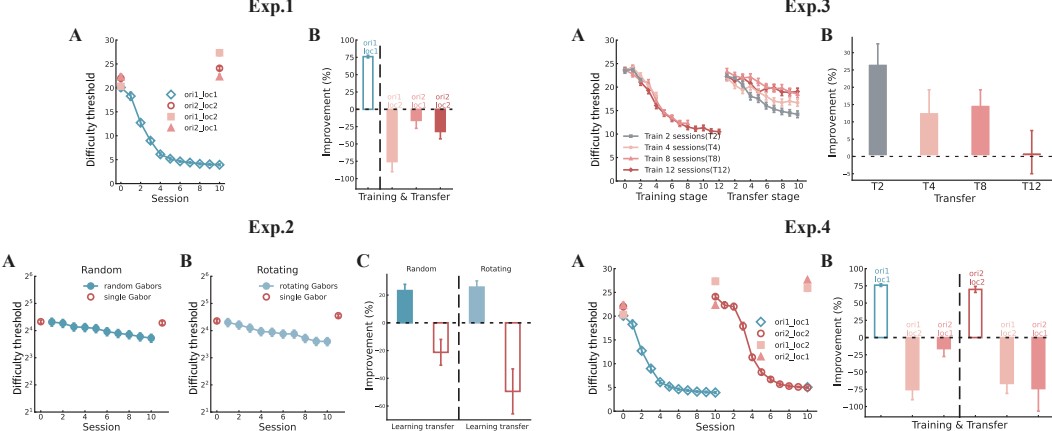

Figure A2: Accelerating feature-based learning results. Repeat the experiments from Sec. 2 with a tenfold increase in the learning rate for feature-based learning. Experiments 1-4 are repeated from those shown in Sec. 2. The results include statistics from 100 simulation experiments, with all other parameters consistent with the model in the main text.

**First Experiment**   Accelerated feature-based learning led to significant modifications in feature representations, resulting in degraded performance at untrained locations (especially for ori1_loc2).

**Second Experiment**   Due to continuous changes in experimental conditions and accelerated learning, the model failed to stabilize representations. As a result, performance in the task was compromised, and human-like transfer was not achieved.

**Third Experiment**   With increased training sessions, differences between conditions diminished, and learning curves became entangled (especially between T8 and T12). The statistical results, shown in Table A5, indicate that the p-values increased compared to Table A4, reflecting reduced significance between conditions.

Table A5: Statistical Comparison of the Third Experiment with Accelerating Feature-based Learning

|      | **T2** | **T4** | **T8** | **T12** |
|------|--------|--------|--------|---------|
| **T2**  | 1       | 0.019 | 4.9e-05 | 5.1e-06 |
| **T4**  | 0.019   | 1     | 0.083   | 0.025   |
| **T8**  | 4.9e-05 | 0.083 | 1       | 0.61    |
| **T12** | 5.1e-06 | 0.025 | 0.61    | 1       |

**Fourth Experiment**   Double training no longer achieves the transfer of learning, and performances in both testing (transfer) conditions are even worse after double training.

### C.4.2 Slowing Down Task-based Learning

As depicted in Fig. A3, we decreased the learning rate of task-based learning by tenfold. This adjustment significantly degraded learning effectiveness, as the model struggled to master tasks efficiently.

**First Experiment**   Although the model could not fully master the task, a comparison with Fig. A2 shows no significant difference in transfer effects between training and transfer conditions.

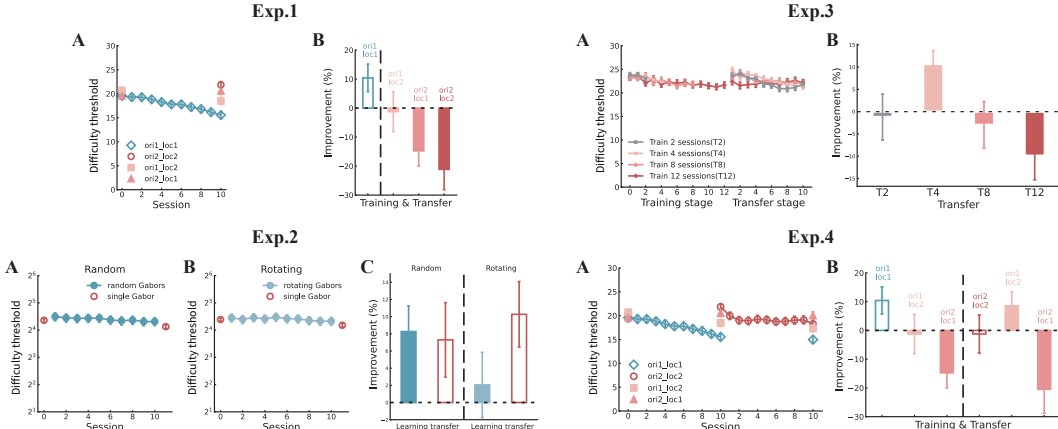

Figure A3: Slowing down task-based learning results. Repeat the experiments from Sec. 2 with the learning rate for task-based learning decreased by a factor of ten. Experiments 1-4 are repeated from those shown in Sec. 2. The results include statistics from 100 simulation experiments, with all other parameters consistent with the model in the main text.

**Second Experiment** Despite the model's poor mastery of tasks, the improvements gained during learning transferred well to new conditions.

**Third Experiment** The slower learning disrupted performance consistency across conditions, leading to entangled learning curves. Statistical analysis, shown in Table A6, reveals increased p-values compared to Table A4, indicating smaller differences between conditions.

Table A6: Statistical Comparison of the Third Experiment with Slowing Down Task-based Learningg

|      | **T2** | **T4** | **T8** | **T12** |
|------|------|------|------|------|
| **T2**  | 1    | 0.68 | 0.97 | 0.76 |
| **T4**  | 0.68 | 1    | 0.71 | 0.48 |
| **T8**  | 0.97 | 0.71 | 1    | 0.73 |
| **T12** | 0.76 | 0.48 | 0.73 | 1    |

**Fourth Experiment** Due to poor learning outcomes, double training failed to achieve transfer. However, a comparison with Fig. A2 shows minimal disruption in transfer conditions.

