# OpenReview forum: "To Learn or Not to Learn, That is the Question — A Feature-Task Dual Learning Model of Perceptual Learning"
_NeurIPS.cc/2024/Conference — NeurIPS 2024 poster_

### Official Review · Reviewer_VB2J · 2024-06-17

**Soundness:** 2
**Presentation:** 2
**Contribution:** 2
**Rating:** 5
**Confidence:** 5

**Summary:**

This paper aims to replicate a variety of results from the perceptual learning literature using a model with two different forms of learning. It shows results that capture how specificity occurs under certain training conditions and transfer occurs under others. The two forms of learning are different in terms of supervision and location of weight changes.

**Strengths:**

The paper tackles an interesting topic, one that is ripe for modeling influence.

Technically they are able to replicate many findings.

**Weaknesses:**

The modeling choices made in this work made it hard to interpret the results and also made the connection to known biology more difficult.  Also there are some issues in the interpretation of the results the authors provide at times.

Specifically:

In the feature-based learning method, the authors tie the weights from different orientations at the same location (this is justified by reference to cortical columns). I don't understand the mechanism that would cause all neurons representing a single location to have the same weight updates. Rather, I would expect there to be orientation-specific learning as orientation is represented differently by different cells.

Different task-based networks are trained for different orientations. The authors do not make this very clear in the main text, but it is of course very important for understanding the results. What is this meant to correspond to biologically? (of course the rest of the ventral stream doesn't change based on which orientation is being used.) The authors also refer to this form of learning as fast, but it includes the learning of many more parameters than the feature-based learning and takes many epochs.

The task-based network also performs convolutions over a space where dimensions represent orientation and space. This means that a downstream cell only gets input from 3 nearby spatial locations and 3 nearby orientations (in the first layer). The spatial specificity is warranted as cells do have spatially restricted receptive fields. But what is the justification for the restricted orientation connections? Furthermore, the fact that the weights are convolutional in this space means that there may actually be orientation transfer in these networks (for the same reason there is location transfer when the task network is trained alone). However this was not tested because different task networks were used for different orientations. The location transfer exhibited by the task-based network also depends on weights being tied (i.e. using a convolution). What is the biological explanation for this feature of the model?

The Hebbian learning (combined with normalization) seems to really mess up the representations at non-trained locations. This is more extreme than simply not transferring to them. How is this interpreted?

To help with interpretation, it would be good if the ablation studies could be run on all the experimental results.

**Questions:**

Is Eqn 2 actually a convolution? It seems like each spatial location has its own separate weight that the activity is multiplied by so I don't see what about it is convolutional. (Also it isn't necessary to include Eqn A4 if the only scenario used is $\alpha=0$).
What values are $A_{t=0}$ initialized as?

The main text makes reference to the task-based network as being a convolutional layer, but it seems like it is actually 3. Also, are there nonlinearities between this layers?

For the third experiment the authors say "At the untrained condition, the threshold first decreases and then increases or saturates, indicating a transition from transfer to specificity". I'm not sure where the increase or saturation is seen. The data is a little noisy but still looks like a continued decrease on average.

I am confused about how the locations of the gabor filters are spread out across the image. Are they overlapping or no? if I just take the width and divide by 40 (as suggested by line 406), the filters would be 20 pixels wide but the standard deviation is 30 so that can't be right. Is there a stride that's missing?

In fig 7, isn't the relative lack of transfer to ori1_loc2 inconsistent with the experimental results? Also 7b makes it look like there is improvement for ori1_loc2 but 7c doesn't. Also it seems the labels are wrong on the right hand side of 7c.

It is possible the authors broke anonymity on line 105 (no need to confirm or deny, just a reminder to be careful)

There are typos in line 145 and 148.

**Limitations:**

The authors should probably speak more to the limitations due to the non-biological components of the model discussed here.

---

> ### Author Rebuttal · Authors · 2024-08-07
>
> Thanks for the very detailed comments.
> We streamline reviewers' main concerns and address them one-by-one.
>
> **Q1:** On the biological plausibility of location-specific plasticity. Based on a number of experimental evidence, we believe that location-specific plasticity is possible in certain conditions.
>
> - First, the Double Training experiments (Xiao et al. 2008, Zhang et al. 2010, Wang et al. 2012, Zhang et al. 2014, Wang et al. 2016, and Xiong et al. 2016 ) and those involving attention to induce transfer (Donovan et al. 2015, 2018, 2020) have indicated that the location-specific plasticity is possible. For instance, Wang et al. (2012) showed that "Double Training" involving stimuli unrelated to orientation, such as distinguishing between a Gabor stimulus and the letter E, could achieve transfer to untrained locations. Xiong et al. (2016) used two experimental paradigms: one in which subjects were presented stimuli without awareness (bottom-up driven), and the other subjects were informed of the presence of stimuli but no such stimuli actually presented (top-down driven), and in both cases, the transfer effect to untrained locations was observed.
>
> - Second, neuroimaging studies have suggested that the results of perceptual learning can transform the processing of location information mediated by top-down signals into those realized by bottom-up implementations. For instance, fMRI findings from Sigman et al. (2005) showed that during the initial stages of learning a visual search task, there was significant activity in the frontal-parietal network, suggesting a pronounced top-level regulatory role; however, as the learning progressed, the frontal-parietal network activity decreased, while the activity in the lower visual cortex increased,
>
> Taken together, we argue that neurons with different orientation preferences can achieve location-specific plasticity through top-down signals (such as attention). The columnar structure can be seen as a way to support position-specific learning.
>
> **Q2:** On the use of different read-out networks. In our simplified model, we only extract features related to angle and position, necessitating the use of different task-based networks to read out different orientation information. We agree that in real biological systems having much more complicated structures, a single network module can process different orientation information. Nevertheless, since the focus of this study is on elucidating the dual-learning framework, in particular, the effect of different speeds of the two learning processes, we could extend the current model to have a single task-based learning module to handle different orientations, which will not change our results qualitatively.
>
> **Q3:** On the use of CNN. The CNN in our model is solely for simulating the task-based learning process with the capability of location generalization. Indeed, there may also be some degree of orientation transfer. Since our study focuses on location specificity, we employed different task-based networks for different orientation discrimination to avoid this interference. It ensures that the learning effect is specific to the location without confusion with orientation changes.
>
> **Q4:** Indeed, the combination of Hebbian learning with normalization can induce distorted feature representations at untrained locations and subsequently cause degraded learning effects at untrained locations. However, this phenomenon is not biologically implausible.
> In real human subject experiments as cited in Fig 1C, it shows that repeated training at specific locations can indeed cause a decrease in subjects' performances at untrained locations.
>
> **Q5:** Equation 2 presents a general learning rule. In practice, for simplicity, we simplified the operation. As shown in Equations A3 and A4, we considered
> $$
> \lim_{a\to0} W_{t+1}(\mathbf{x}, \mathbf{x}^\prime) = \frac{A_{t+1}(\mathbf{x}^\prime)}{\sqrt{2 \pi} a} \exp \left[-\frac{(\mathbf{x} - \mathbf{x}^\prime)^2}{2 a^2}\right],
> $$
> which shows that feature-based learning employs a Gaussian kernel to perform weighted convolution on representations. For simplicity, we considered the scenario where parameter $a$ approaches zero. In this case, significant weight updates occur only when the input and output positions are exactly the same, effectively creating the one-to-one connection between neurons at identical locations.
>
> **Q6:** See reply to Q4 of the Reviewer UGym.
>
> **Q7:** See reply to Q11 of the Reviewer Wypk.
>
> **Q8:** Apologies for the typo. It should be "divide by 10".
>
> **Q9:** Apologies for the confusion, indeed the labeling of "loc1\_ori2" and "loc2\_ori1" in Fig.7b were reversed. After correcting the labeling, it can be seen that our results are consistent with the experimental results in Xiao et al. 2008 (Fig.3).
>
> After re-evaluating the improvement results shown in Fig.7b and Fig.7c, we find that the inconsistency arises due to that Fig.7b presented the average of 100 simulation runs, while the improvement in Fig.7c was calculated by averaging the rate of improvement from each simulation, where a few outliers affected the results. After excluding 4 outliers, the results are consistent, see Fig.3 in the attached PDF.
>
> **Q10:** Thanks for the reminder.
>
> **Q11:** Typos will be corrected.

---

> > ### Comment · Reviewer_VB2J · 2024-08-12
> >
> > I thank the authors for their response.
> > With respect to the answer to Q1: The behavioral studies that show location transfer is possible do not directly support this specific mechanism of location transfer. That would require evidence on the neurophysiological level. I believe the second point is trying to say that changes in top-down modulatory influence might be responsible for aiding location transfer (but the mechanism is vague and not implemented here). In any case, the authors should acknowledge this modeling gap more directly in the main text so as to not mislead the reader about the mechanisms of the model. Furthermore, it will help explain some trends in results such as the slight transfer to orientation 2 at location 1 seen in Figure 4B.
> >
> > WRT Q 2&3: The authors say "Since our study focuses on location specificity", but the study is presenting results about orientation transfer as well.  So it does seem important to reflect on how the choice of readout model/task-based learning influences that process. To even test orientation transfer requires several epochs of training of a different readout model; the details of this model/training process could influence the results. It is also said that "a single task-based learning module to handle different orientations [..] will not change our results qualitatively." My point is that it could, depending on how it is built. That is why I'm advocating for more transparency in the main text about how the model is built, how transfer is tested, and how these design choices might impact or explain the results. I think a reader would benefit from the author's careful reflection on these issues.
> >
> > Q4: I looked at reference 24 and I don't see much evidence of extended training causing a decrease in performance at the transfer location. There were maybe 1 or 2 individual subjects where the average discrimination threshold on the first transfer session was slightly higher than the threshold on the first training session, but I'm not clear if these were even statistically significant differences. The more common trend however was a slight decrease in threshold on the first transfer sessions (i.e. partial transfer, even for extended training).

---

> > > ### Author Response · Authors · 2024-08-13
> > >
> > > Thank you for the instructive comments.
> > >
> > > - Regarding question 1: We agree that the detailed neural mechanisms for perceptual learning in the brain remain unclear. The goal of this study is not to assert that the nervous system actually employs the models/learning algorithms we used in this work, but rather to propose a plausible explanatory framework for perceptual learning. As suggested by the reviewer, in the revised manuscript, we will discuss in much more detail about the biological plausibility of the models/learning algorithms to clarify the underlying assumptions.
> > >
> > > - Regarding questions 2 & 3: Thanks for the suggestion. We will further clarify the settings of the models in the paper to avoid potential misunderstandings. Our previous statement that "a single task-based learning module handling different directions [...] would not qualitatively change our results" is based on the current structure and settings of our model.
> > >
> > > - Regarding question 4: Thanks for the comments. We should have made our points more clearly. In the paper [24], the authors found that with more training in the first task, the initial threshold in the new task became higher, and so did the final achieved threshold (thus the learning performance was decreased compared to the less trained case). This indicates that extensive training in the first task can alternate neural representation, increasing the difficulty of learning the new task (as shown in our Fig1C; and Fig2 in [24], comparing the training of two and twelve blocks, T2 black and T12 green curves).

---

> > > > ### Comment · Reviewer_VB2J · 2024-08-13
> > > >
> > > > Thanks for the response. Under the assumption that the authors will make the stated changes to better explain the model mechanisms, I have increased my score.

---

> > > > > ### Author Response · Authors · 2024-08-14
> > > > >
> > > > > Thank you for your positive remarks. We will incorporate your suggestions in the paper. Your suggestions have greatly improved our paper, please let us know if you have other questions or concerns.

---

### Official Review · Reviewer_Wypk · 2024-07-02

**Soundness:** 3
**Presentation:** 4
**Contribution:** 4
**Rating:** 7
**Confidence:** 4

**Summary:**

In this article, the authors propose a novel model that accounts for two different phenomena observed in human learning: i) specificity, a feature-based mechanism restricted to the very specific statistics of the environment condition, and ii) transfer, a task-based mechanism that allows to transfer knowledge to untrained locations or features. The model proposed by the author is made of a stack of 3 components: A feature extraction stage (mimicking LGN), a Feature-based learning stage (mimicking early processing), and a task-based learning (mimicking upper-processing). Training this model on a vernier discrimination task, the authors demonstrated that it accounts for numerous psychophysics phenomena: i) specificity in condition-specific training, ii) transfer in various training conditions, and ii) transition from transfer to specificity when trained on an increased number of training sessions.

**Strengths:**

This article is well written, is well motivated, and leverages simple, yet very informative, experiments that could be easily compared to human data. Overall it was pleasant to read. Great work!

**Weaknesses:**

I have noted various (rather minor) points that needed to be better clarified/explained (see question section). My only concern is about the generalization of the proposed model to more complex tasks (including for example natural image discrimination). I fully agree that simple tasks like the Vernier discrimination are a very good starting point to propose a novel model accounting for human learning, but scaling such models to more complex data would have drastically improved the impact of this article. Overall I don’t think this scaling-up argument is enough to reject this article, but this is something the authors need to keep in mind if they want their article to reach a more ‘global’ audience. Note that I am willing to increase my rating if the points (in the question section) are properly addressed/clarified.

**Questions:**

1 - In my understanding, feature-based is supposed to account for selectivity, which is thought to happen when over-training is taking place (i.e. when stimuli are relatively similar, and there that the environment is relatively unchanged). But it seems to contradict this sentence (l75-77): ‘Feature-based learning … only takes place when the external environment is substantially changed ». On my understanding if the environment is « substantially changed » then we are not in the over-training condition... Am I missing something? Could you explain?

2 - In Figure 2A, this is not clear what loc(x) corresponds to ? Is it encoded as the x,y coordinate of the stimulus? Please give more detail.

3 - In Appendix 1 (Eq.A4), you suggest a=0, which involves two divisions by zero (one in the exponential multiplier, and another inside the exponential). Is that a typo or is that a=0? If this is the case how do you handle the division by zero?

4 - What is the grey line in Figure 3?

5 - In Fig 3.C. How do you explain that the accuracy for the training sample is decreasing with only feature-based learning? I would expect to see this accuracy improving if the feature-based learning model well the specificity phenomena as we observe an increase in performance for human data (Fig 1Aii ).

6 - I am not sure the claim line 173 is true: « feature based learning… reinforces the model’s performance at the trained location ». This is the opposite trend that is observed in Fig 3C… Could you discuss this point?

7 - In Fig 3D: There is no mention of the method used to compute the similarity in Fig 3D. Could you elaborate?

8 - Line 182: «  The combination of feature-based and task-based learning accelerates the training process at the train location ». This is not visually obvious when we compare Fig 3B and 3E… Could you quantify the difference in convergence speed? Ideally, make sure this difference is statistically significant.

9 - Could you explain a bit more about the difficulty threshold in section 4? And what is the difference between the discrimination threshold and the difficulty threshold? Is that the same? If I understood well the difficulty threshold is the offset between the two Gabor filters (the smaller the offset the harder the task). But in line 191, you introduce the discrimination threshold, which seems to be more related the the intensity of the stimuli than the offset. Could you clarify this point?

10 - Figure 6: Could run a statistical test to make sure the T4 is indeed below T8 (in Fig6b, in the transfer setting?

11 - Not sure what you mean in line 218: In the untrained condition, the threshold first decreases and then increases or saturates… I am not sure to see that in Fig 6B… could please be more accurate here

### Typo :
* Fig2C : learining —> learning
* Line 144: Sec. A.1 —> Sec A.2

**Limitations:**

The author properly discusses the limitations of their paper.

---

> ### Author Rebuttal · Authors · 2024-08-07
>
> Thanks for the careful and valuable comments.
> We streamline reviewers' main concerns and address them one-by-one.
>
> **Q1:** On the goal of feature-based learning. In this study, we argue that the goal of feature-based learning is to capture the statistical characteristics of external features. Therefore, it only takes effect when there is a significant change in the distribution of external features. In our experiments, when the stimuli are presented repeatedly at the same location for many times,
> triggering the sense of statistical change of external stimuli by the brain,
> the weight changes associated with feature-based learning become pronounced, eventually dominating the model's performance and demonstrating specificity. This mechanism highlights how feature-based learning adapts to repetitive patterns, enhancing the model’s ability to specialize in recognizing consistent (salient) features in its environment.
>
> **Q2:** Apologies for the confusion. Indeed, $\mathbf{x}$ should be a vector indicating the position. Specifically, for the task at hand, $loc(\mathbf{x}) $represents the coordinates as $\mathbf{x}=(x,y)$, where $x$ and $y$ denote the horizontal and vertical coordinates, respectively.
>
> **Q3:** Apologies for the confusion. Here, we actually consider the case of limit. Specifically:
> $$
> \lim_{a \to 0} W_{t+1}(\mathbf{x}, \mathbf{x}^\prime) = \frac{A_{t+1}(\mathbf{x}^\prime)}{\sqrt{2 \pi} a} \exp \left[-\frac{(\mathbf{x} - \mathbf{x}^\prime)^2}{2 a^2}\right].
> $$
> As the width $ a $ approaches zero, the Gaussian kernel becomes very sharp, indicating that weight updates are concentrated at $ \mathbf{x}=\mathbf{x}^\prime $. In this limit scenario, significant weight updates occur only when the input and output positions are exactly the same, corresponding to the one-to-one connection.
>
> **Q4 and Q8:** In Fig.3B and Fig.3E, the gray line indicates the number of training epochs required for the model to achieve a 90\% accuracy rate. This metric serves as the reference point for assessing the convergence speed of the model under different learning conditions.
>
> Under the task-based learning only condition (as shown in Figure 3B), the model requires approximately 75 training epochs to reach 90\% accuracy.
> However, under the condition of combining task-based and feature-based learning (as shown in Figure 3E), the model converges faster, needing only about 68 training epochs to achieve the same level of accuracy.
>
> A statistical significance was found between these two conditions, with a p-value of approximately 0.00148 (< 0.005) in one hundred simulations.
>
> **Q5-7:** Questions 5 to 7 are closely related. Feature-based learning is independent of task requirements, so it is possible that it may reduce the model's performance. For human subjects, feature-based learning and task-based learning occur simultaneously, thus they cannot be compared separately. This explains why we do not observe enhanced feature processing at the trained locations in Fig.3C. Rather, we should focus on Fig.3D, where we assess similarity by calculating the correlation between the representations before and after feature-based learning. The correlation is calculated as follows:
> $$
> corr = \frac{n \left(\sum F^*_t F_t \right) - \left(\sum F^*_t \right)\left(\sum F_t \right)}{\sqrt{\left[n \sum (F^*_t)^2 - \left(\sum F^*_t \right)^2\right] \left[n \sum (F_t)^2 - \left(\sum F_t \right)^2\right]}}.
> $$
> Here, $ F^*_t $ and $ F_t $ represent the representations before and after the feature-based learning module, respectively, and $ n $ is the total number of representations.
>
> **Q9:** In Section 4, all our model simulations utilize difficulty thresholds. Specifically, we adjusted the offset between two vertical Gabor filters in the task to create a series of difficulty levels, and we selected the difficulty level that corresponds to 80\% accuracy of the model as the threshold (mentioned in line 193 and Appendix C2). The "discrimination threshold" introduced in line 191, however, refers to the threshold used in psychophysical experiments to assess learning performance.
> Although these two types of thresholds differ, their roles are similar: they are both used to measure the limit at which the model achieves a certain performance under specific conditions.
>
> **Q10:** The results of the t-tests conducted on the data from Fig.6 are presented below, highlighting the comparison between T4 and T8. As observed, the statistical differences between these two conditions are not significant. This finding aligns with the results depicted in Fig.1C(ii), indicating the consistency across different parts of the study.
>
> |     | T2                 | T4                 | T8                 |T12                 |
> |-----|-------------------|-------------------|--------------------|--------------------|
> | **T2** | 1                 | 0.025  | 4.1e-05            | 3e-07 |
> | **T4** | 0.025             | 1      | 0.075              |0.0018|
> | **T8** | 4.1e-05           | 0.075  | 1                  |0.11|
> | **T12** | 3e-07             | 0.0018 | 0.11               |1|
>
> Table 1: Statistical comparison of Fig.6
>
> **Q11:** We revised the description as follows: "The results are presented in Fig.6B, which show that the thresholds decrease gradually in the trained condition (left panel in Fig.6B). In conjunction with Fig.3E, it can be observed that in the untrained condition, the threshold should first decrease and then either increase or saturate. This indicates that with increased training, there is a transition from transfer to specificity at untrained locations." This describes how training intensity influences learning dynamics differently in trained versus untrained locations.

---

> > ### Comment · Reviewer_Wypk · 2024-08-12
> > **response to author**
> >
> > I am convinced by the author's rebuttal. I increase my rating to 7

---

> > > ### Author Response · Authors · 2024-08-12
> > >
> > > Thank you for the improved score. We are grateful for the valuable suggestions you provided, which have enhanced the clarity of our work. We will incorporate these insights into our revised manuscript.

---

### Official Review · Reviewer_n6Xq · 2024-07-05

**Soundness:** 3
**Presentation:** 3
**Contribution:** 2
**Rating:** 5
**Confidence:** 2

**Summary:**

1. The paper proposes a dual-learning model to reconcile two seemingly contradictory phenomena in perceptual learning: specificity and transfer.
2. The model consists of two learning processes:
   - Task-based learning: Fast, enables quick adaptation to new tasks using existing neural representations.
   - Feature-based learning: Slow, refines neural representations to reflect changes in the environment.
3. The model is implemented as a hierarchical neural network with three stages:
   - Feature extraction
   - Feature-based learning
   - Task-based learning
4. The interactions between these two learning processes explain the observed specificity and transfer effects in perceptual learning experiments:
   - Specificity occurs when feature-based learning dominates (due to excessive training on the same stimulus).
   - Transfer occurs when task-based learning dominates (due to varied training conditions).
5. The model successfully reproduces key experimental findings in perceptual learning, including:
   - Specificity in condition-specific training
   - Transfer in varied training conditions
   - Transition from transfer to specificity with increased training sessions
   - Transfer effects in double training paradigms

**Strengths:**

The paper proposes a simple, novel dual-learning model that effectively reconciles the conflicting phenomena of specificity and transfer in perceptual learning. With the help of this model, the authors successfully reproduces classical findings from perceptual learning experiments. The paper is easy to follow and experiments seem sound.

**Weaknesses:**

I am not a computational neuroscientist and only follow this field very sparsely. I found the paper interesting but can not judge the novelty of the paper and its methodology and results. Nevertheless, I would argue that the experimental section is missing ablations or theoretical results to explain the (many) moving bits and pieces , and therefore authors design choices, of the proposed model.

**Questions:**

Please ablate the model.
1) Is the first hand designed feature extractor necessary? What happens you mess this part up?
2) What happens if you train  the middle part with backprop?
3) What happens if you train always the first and the last part with backprop simultaneously.

Generally I think the proposed model seems quite ad how and potentially makes intuitive sense. Please back up design choices with experimental results. Show me when model behaviors start misalignment with the expected findings of perceptual learning.

**Limitations:**

Seems ok to me.

---

> ### Author Rebuttal · Authors · 2024-08-07
>
> Thanks for the valuable comments of the reviewer.
>  We streamline reviewers' main concerns and address them one-by-one.
>
> **Q1:**  On the removal of the feature-extraction module.
> The feature extraction module is akin to a vision representation extractor that has been trained through extensive experiences over time. If we remove or disrupt this module, other parts of the model will not be able to operate effectively, rendering the entire model ineffective.
>
> **Q2:**  On the choice of Hebbian learning or backpropagation.
> In the dual-learning framework, feature-based learning
> aims to capture the distribution variation of external stimuli, which is task-independent and requires a significant number of observations. Therefore, we used slow and unsupervised Hebbian learning
> to achieve this goal.
> In contrast, task-based learning is directly linked to task execution using the existing representations, and it can be quickly achieved by modifying the read-out weights. We therefore chose the fast and supervised BP to implement this part. Overall, the choice of the learning method is based on the biological plausibility.
>
> **Q3:**  The newly added ablation studies.
> We add two ablation studies to highlight the importance of the relative speeds of two learning processes.
>
> **1. Accelerating feature-based learning.** As shown in Fig.1 of the attached PDF, we increase the learning rate of feature-based learning by tenfold and replicate the four experiments in the paper. The results are:
>
> - **In  Exp1**, the results do not differ significantly from the original model, but the accelerated feature learning significantly modifies feature representations, leading to significantly degraded performances at untrained locations (ori1\_loc2).
> - **In Exp2**, due to continuously changing experimental conditions and the accelerated feature learning rate, the representations are constantly altered, preventing the model from performing well in the learning task and failing to achieve human-like transfer.
> - **In Exp3**, the transferability of learning in new conditions diminishes with increasing training numbers, but with significant changes in feature-based learning, the differences in learning curves under new conditions are no longer distinct and become entangled, showing a decreasing trend in learning gain (except for T12), with increased error bars.  However, when these differences are statistically analyzed using a T-test compared to the results in the main text (as detailed in the reply to the Reviewer Wypk), the p-values have increased, suggesting reduced statistical significance between conditions.
>   |     | T2    | T4    | T8    | T12   |
>   |-----|-------|-------|-------|-------|
>   | T2  | 1   | 0.049| 0.00088| 0.12 |
>   | T4  | 0.049  | 1      | 0.29     |0.5|
>   | T8  |0.00088  | 0.29   | 1 |0.053|
>   | T12 | 0.12 | 0.5 | 0.053       |   1|
>
> Table 1: Statistical comparison of Exp3 with accelerating feature-based learning
>
> - **In Exp4**, double training no longer achieves the transfer of learning, and performances in both transfer conditions are even worse after double training.
>
> **2. Slowing down task-based learning:** As depicted in Fig.2 of the attached PDF, we decreased the learning rate of task-based learning by tenfold. As expected, this adjustment resulted in a degradation of the learning effect, rendering the model less capable of mastering the tasks efficiently.
>
> - **In Exp1**, although the model cannot fully master the current task, a comparison with Exp1 in Fig.1 reveals that there is no significant difference in the effects of training on other conditions.
>
> - **In Exp2**, especially under random conditions, it is observed that despite the model's inability to master the current task effectively, the improvements brought about by learning still transfer to new conditions.
>
> - **In Exp3**, due to the model's inability to effectively master the current task, the learning curves for all four different conditions are almost entangled. Significance testing shows that p-values have increased, indicating smaller differences between different training conditions:
>
>   |     | T2    | T4    | T8    | T12   |
>   |-----|-------|-------|-------|-------|
>   | T2  | 1     | 0.68  | 0.089 | 0.94  |
>   | T4  | 0.68  | 1     | 0.02  | 0.76  |
>   | T8  | 0.089 | 0.02  | 1     | 0.095 |
>   | T12 | 0.94  | 0.76  | 0.095 | 1     |
> Table 2: Statistical comparison of Exp3 with slowing down task-based learning
>
> - **In Exp4**, due to poor learning outcomes, double training did not facilitate the transfer of learning. However, when compared with Exp4 in Fig.1, it can be seen that there is minimal disruption to transfer conditions.
>
> Due to time constraints, the above ablation study results are only statistical outcomes from 20 simulations. We plan to conduct 100 experimental simulations consistent with the main text and will perform a more detailed analysis. However, we do not expect significant deviations from the current ablation study results.
>
> These two sets of ablation studies demonstrate that both slow feature-based learning and fast task-based learning are necessary for our model to reproduce perceptual learning phenomena.

---

> > ### Comment · Reviewer_n6Xq · 2024-08-11
> > **Thank you!**
> >
> > Thank you for the additional clarification and data. I will raise my score to 5, but highlight my lack of knowledge wrt the biological plausibility and relevance of the algorithm for neuroscienctists - so I can not judge the contribution of the paper fairly.

---

> > > ### Author Response · Authors · 2024-08-12
> > >
> > > Thank you for the score bump. We are pleased that the ablation study you suggested has enhanced the clarity of our work, and we will include these changes in our revised manuscript.

---

### Official Review · Reviewer_UGym · 2024-07-13

**Soundness:** 4
**Presentation:** 4
**Contribution:** 3
**Rating:** 8
**Confidence:** 4

**Summary:**

The paper puts forth a theoretical framework for perceptual learning, in which two separate learning processes contribute to learning a perceptual task (a fast, flexible task-based learning that relies on existing feature representations; and a slow, task-specific feature learning). Repeated learning sessions with the same stimulus conditions triggers feature-based learning, which will be specific to the training stimulus conditions. The framework is instantiated as a neural network model, and is used to account to both specific and transfer phenomena in psychophysical experiments.

**Strengths:**

The paper proposes a novel computational model to explain when specificity and transfer is observed in perceptual learning. The crux of the proposed dual-learning theory is that experiments exhibiting transfer are dominated by task-based learning (which operates over existing feature representations and is adaptable), and that experiments exhibiting specificity are dominated by feature-based learning triggered by excessive exposure to the same stimulus condition triggering feature-learning to adapt to these new environment statistics.

The computational framework is instantiated as a neural network model with feature extraction (via a set of basis functions), feature-based learning which modifies the feature representations via hebbian learning, and a task network (CNN that performs the task). This network recapitulates an impressive array of human psychophysical phenomena, showing specificity when repetition of the same stimulus conditions is high, and transfer when there is more variability in the stimulus conditions. The framework impressively captures nuances of learning dynamics across multiple task variations.

As such, the work provides a strong framework for understanding diverse and seemingly contradictory findings in human psychophysics. It also presents an opportunity for impact on machine learning theory, if the slow-feature-learning and fast-task-learning framework has broader benefits for machine learning systems (e.g., could this inform approaches to life-long continual learning research?).

**Weaknesses:**

One important limitation of this work is that the networks appear to be trained from scratch on the “probe task” (e.g., vernier acuity). In contrast, with human perception, the person’s visual system is tuned over their lifetime (presumably via some self-supervised objective), and then they are put into an experimental setting where 1-2 hours per day for N days the perform this new task. So the impact in terms of perceptual experience is set within this very broad context of “learning to see”, and a visual system that’s capable of much more than just the task at hand (e.g., vernier acuity). I suppose this begs the question of whether this all works as well if the network is initially pretrained (say on some self-supervised learning objective), and then the same types of experiments are run. I don’t see in principle why these same ideas wouldn’t apply.

Minor: When the computational framework is first introduced, the different architectural choices are unmotivated (a classical basis function network for image processing — What is this and why this? A Hebbian network for feature learning – Why a Hebbian network? Why is feature learning a separate step from the basis function learning? And then a CNN with max pooling for task-based learning. Why a CNN for this purpose, and a Hebbian network for the feature learning?). I realize these choices are all motivated, they just come out of the blue all at once, and it’s hard for the reader to grok the motivation for this setup.

Minor: The methods introduce the Feature Extraction as akin to the retina or LGN, but what’s described appears to be an orientation-tuned Gabor Wavelet Basis Set (and orientation-tuning is believe to arise later in V1 and beyond).

Minor: Can you provide more information on the architecture of the task-based CNN?

Minor: Can the authors provide further justification/clarification for the different learning algorithms implemented at each level of the network? e.g., Why shouldn’t Hebbian learning play a role “all the way to the top” of the task network? Is this biologically motivated, or a convenient way to implement assumptions of the model (e.g., location-specific learning)?

Minor: Other related work includes “How variability shapes learning and generalization” (Raviv et al, Trends in Cognitive Sciences)

**Questions:**

Can the authors clarify what makes the task-based learning “fast”? Is this accomplished via learning rate hyperparameter tuning?

**Limitations:**

OK

---

> ### Author Rebuttal · Authors · 2024-08-07
>
> Thanks for your encouraging and valuable comments. We streamline reviewers' main concerns and address them one-by-one.
>
> **Q1:** In the current study, we have used a feature extraction module that remains unchanged during the learning process to reflect the pre-processing of visual inputs in the brain. This setup mimics the resulting capability of the human visual system that has undergone extensive exposure to diverse environments throughout life. In the typical psychophysical experiments which last at most hours, we regard this part of pre-processing as unchanged. To accomplish the Vernier discrimination task in this work, we have employed a simple basis function network for pre-processing. Certainly, if more complicated tasks are included, we can employ more sophisticated pre-trained models.
>
> **Q2:**  As for the choice of models and learning methods, please refer to the overall rebuttal.
>
> **Q3:**  Indeed, as pointed out by the reviewer, orientation tuning emerges in the visual cortex. Our description of feature extraction in the paper was not precise. The feature extraction module in our model should include the input layer of V1 (responsible for extracting visual features). We will revise the statement in the manuscript accordingly.
>
> **Q4:**  The CNN used in our model is a streamlined convolutional neural network comprising three convolutional layers.
> - The initial layer, layer 1, inputs a single channel and uses a 3x3 convolutional kernel to output 6 channels.
> - This is followed by layer 2, which processes these 6 channels through another 3x3 kernel to produce 10 channels.
> - The concluding layer, layer 3, compresses these 10 channels into a single output channel using a 3x3 convolution.
>
> The network employs ReLU activation after the first two convolutional layers to add non-linearity and includes a sequence of flattening and a 1D max pooling on the final output to structure the output appropriately.
>
> **Q5:**  As for the model choice, please refer to our overall rebuttal. Indeed, the most fundamental difference lies in the objective functions of the two learning processes. Utilizing Hebbian learning (Hebb) or backpropagation (BP) mainly simplifies the model implementation.
>
> **Q6:**  From the perspective of the dual-learning framework, variability enhances generalizability primarily because it prevents feature-based learning from confining learning effects to specific feature combinations, thus allowing task-based learning to dominate and exhibit transferability.
>
> On the other hand, specificity also plays an important role in learning. For instance, according to Reicher's study in 1969, native English speakers, due to extensive reading in English, can recognize words significantly faster than individual letters or strings that do not conform to phonetic rules. This indicates that prolonged training on specific types of inputs can significantly enhance the efficiency and accuracy of processing these inputs.
>
> Overall, our model aims to simulate these learning dynamics observed in real-world scenarios. It emphasizes the interaction between feature-based and task-based learning, highlighting their joint effects on perceptual recognition and task execution capabilities. Both learning are valuable for the brain to adapt to and interact with complex environments.
>
> **Q7:** Yes, we set a much higher learning rate for task-based learning, resulting in the network parameters related to task-based learning being updated much more quickly, which allows the model to rapidly adapt to the task replying on the existing feature representations.

---

### Author Rebuttal · Authors · 2024-08-07

We acknowledge the very careful and valuable comments of all reviewers. We realize that there are common concerns about the aim of this study and the models we used to demonstrate the framework. In the below, we briefly summarize the motivation and main results of this work to clarify these concerns.

Overall, in this work, we explored the neural mechanisms underlying perceptual learning and proposed a dual-learning framework in which the interplay between **the rapid, supervised, task-based learning** and **the slow, unsupervised, feature-based learning** generates the rich phenomena of perceptual learning. Our model reconciles the seemingly conflicting phenomena of specificity and transfer observed in diverse experiments.
To our knowledge, our work is the first one that uses a computational model to elucidate the dual-learning framework for perceptual learning.

Since our focus is on elucidating the dual-learning framework, we have chosen as simple as possible, and meanwhile, as biologically plausible as possible, models to implement each part of learning. Nevertheless, if other models can capture the characteristics of one part of dual-learning, they can be used in our framework.
- Specifically, in our model, the feature extraction module is dedicated to transforming images into visual features, representing a stable representation system in the brain that rarely changes. To accomplish the visual discrimination task in this work, we employed the classical basis function network. Certainly, if more complicated tasks are considered, more sophisticated pre-trained models can be used.
- The feature-based learning module aims at learning the variation in the statistical distribution of visual inputs. It takes effect only when significant changes have occurred in the distribution of visual inputs, and hence it is relatively slow and goes in an unsupervised manner. We employed the classical Hebbian learning with a slow learning rate to implement feature-based learning.
- The task-based learning module is driven by the task at hand, aiming to extract task-relevant information from existing representations to accomplish the task rapidly. Hence, we used a CNN model with supervised learning (such as backpropagation) having a relatively large learning rate to implement this learning process.

To highlight the importance of the interplay between
the rapid task-based learning and the slow feature-based learning, we have carried out an additional ablation study (see the attached pdf Fig.1 and Fig.2),
which varies the learning rates of two learning processes, and demonstrates that the relative speed ratio between two learning processes is essential for perceptual learning.

---

### Decision · Program_Chairs · 2024-09-25

**Decision:**

Accept (poster)

**Comment:**

In this study, the authors proposed a dual-learning framework in which the interplay between the rapid, supervised, task-based learning and the slow, unsupervised, feature-based learning to explain a wide range of phenomena in perceptual learning. Although the reviewers had several concerns about the motivation/model design/Hebbian learning mechanisms, the rebuttal successfully convinced the reviewers. I am inclined to accept the paper.